# Evaluating practical support stroke survivors get with medicines and unmet needs in primary care: a survey

James Jamison,[1] Luis Ayerbe,[2] Gian Luca Di Tanna,[2] Stephen Sutton,[1] Jonathan Mant,[1] Anna De Simoni[2]

[1]Primary Care Unit, Department of Public Health and Primary Care, University of Cambridge, School of Clinical Medicine, Cambridge, UK
[2]Centre for Primary Care and Public Health, Barts, The London School of Medicine and Dentistry, London, UK

**Correspondence to**
James Jamison;
jj285@medschl.cam.ac.uk

## ABSTRACT

**Objectives** To design a questionnaire and use it to explore unmet needs with practical aspects of medicine taking after stroke, predictors of medicine taking and to estimate the proportion of survivors who get support with daily medication taking.

**Design** Four workshops with stroke survivors and caregivers to design the questionnaire. A cross-sectional postal questionnaire in primary care.

**Setting** 18 general practitioner practices in the East of England and London. Questionnaires posted between September 2016 and February 2017.

**Participants** 1687 stroke survivors living in the community outside institutional long-term care.

**Primary outcome measures** The proportion of community stroke survivors receiving support from caregivers for practical aspects of medicine taking; the proportion with unmet needs in this respect; the predictors of experiencing unmet needs and missing taking medications.

**Results** A five-item questionnaire was developed to cover the different aspects of medicine taking. 596/1687 (35%) questionnaires were returned. 56% reported getting help in at least one aspect of taking medication and 11% needing more help. 35% reported missing taking their medicines. Unmet needs were associated with receiving help with medications (OR 5.9, P<0.001), being on a higher number of medications (OR 1.2, P<0.001) and being dependent for activities of daily living (OR 4.9, P=0.001). Missing medication was associated with having unmet needs (OR 5.3, P<0.001), receiving help with medications (OR 2.1, P<0.001), being on a higher number of medicines (OR 1.1, P=0.008) and being older than 70 years (OR 0.6, P=0.006).

**Conclusions** More than half of patients who replied needed help with taking medication, and 1 in 10 had unmet needs in this regard. Stroke survivors dependent on others have more unmet needs, are more likely to miss medicines and might benefit from focused clinical and research attention. Novel primary care interventions focusing on the practicalities of taking medicines are warranted.

## INTRODUCTION

Stroke is the leading cause of disability in developed countries, with an estimated 25%–74% of the 50 million stroke survivors

## Strengths and limitations of this study

► Development of the questionnaire was based on patients' and caregivers' own views gathered through workshops.
► Stroke survivors were recruited from two UK regions.
► This work identified issues from a population that includes patients severely affected by stroke, who are often excluded from research.
► Results shed light on the effect of stroke-related impairments on practical domains and predictors of medicine taking, which have significant effects on medication adherence and call for new primary care interventions.
► The low response rate reported is a limitation of this study, and stroke survivors who are harder to reach may have been missed.

worldwide requiring some assistance or being fully dependent on caregivers for activities of daily living (ADLs).[1–3] For many older adults remaining independent at home may depend on how well they can manage complex medication regimens.[4 5] Around half of stroke survivors are dependent on others for everyday activities.[6]

There is evidence that being dependent for ADLs and impairment in mobility and communication decrease medication adherence in patients suffering from hypertension.[7] Deficits in attention, cognition or working memory have been linked with non-adherence to medications in other patient groups.[8] In a recent systematic review of medication adherence among patients with cognitive impairment, one-third of studies showed that such patients were likely to have a caregiver to assist with medications and there was an association between taking four or more medicines and non-adherence.[9] In patients taking cardiovascular medicines, multiple factors including cognitive problems, lack of social support, dosing regimen, as well as practical problems and difficulties accessing services, contribute to poor medication adherence.[10 11]

Low adherence to secondary prevention medication is associated with poor cardiovascular health.[12 13]

Stroke survivors have previously reported difficulties in the handling of medication as a barrier to adherence to secondary prevention medication after stroke.[14] This was true irrespective of age at stroke, with younger and older stroke survivors being similarly affected.[14] Research on medication adherence in stroke has identified multiple barriers to medication taking among stroke survivors.[14–16] However, interventions developed to improve adherence have mainly concentrated on patients responsible for their own medicine taking.[17 18]

In England, the average age at stroke is 74 for men and 80 years for women.[19] In elderly patients in particular, cognitive deficits, taking large number of medicines and the complexity of medication regimens have been identified as barriers to medication adherence.[20 21] Caregivers are known to play a key role in providing assistance to older people in a range of daily activities including medication taking and physician visits[22] and can help improve adherence in cardiac patients with memory problems.[23]

Survivors of stroke have previously reported unmet needs including physical difficulties, cognitive and emotional difficulties, information needs and other unmet needs.[24 25] However, we know little about factors that influence medication taking among stroke survivors with disabilities (ie, physical or cognitive) living in the community (ie, not in nursing homes), their unmet needs around the use of medicines or the proportion relying on caregivers for some or all aspects of medicine taking.

To date, survey instruments examining the unmet needs of stroke survivors have not focused on practical aspects of medication taking such as how patients collect or handle their medicines.

The aims of this investigation were to design an instrument to evaluate the support stroke survivors get with taking their medicines, characterise patients receiving help with medications, estimate the proportion who have unmet needs with daily medicine taking and who miss medications. We additionally aimed to identify the predictors of missing medicines and of experiencing unmet needs with medications.

This knowledge can inform the development of primary care interventions aimed at improving medication taking in this patient group.

## METHODS
### Questionnaire development workshops
To develop the questionnaire, current literature evidence was evaluated[17] and 3 workshops were conducted with 26 stroke survivors and 12 caregivers in the East of England (St John's College, Cambridge 2009[26]: 7 patients, 1 caregiver; Different Strokes, Cambridge 2012: 9 patients, 3 caregivers; Peterborough, 2012: 10 patients, 8 caregivers). Recruitment was opportunistic and no purposive sampling was applied. The workshops were organised in the context of gathering patient and public involvement

(PPI) input into research grant applications aimed at improving adherence to medication after stroke.[17]

The survey questions were developed through thematic analysis[27] of workshops field notes.

A fourth workshop was conducted to gather feedback on the questionnaire using a PPI exercise with 11 stroke survivors and 3 caregivers recruited through a local stroke group (Different Strokes, East of England). Two stroke survivors from this group took part in subsequent 'think-aloud' interviews, which involved talking out loud as they read the questionnaire, continually verbalising what they were thinking.

### Postal survey
In respect to sample size, 400 returned questionnaires would allow good precision for prevalence estimates. The 95% CIs on various proportions with this sample size were calculated using the Wilson score method (with continuity correction) and are as follows: 50% (45.00% to 55.00%), 25% (20.89% to 29.60%), 5% (3.16 to 7.74%). With 600 questionnaires, the improvement in the precision of the estimates would be as follows: 45.93% to 54.07%, 21.62% to 28.70% and 3.46% to 7.14%, respectively.

General practices in primary care in the East of England and London were approached through the Clinical Research Network (CRN). CRN Eastern contacted 20 general practitioner (GP) practices, of which 11 replied and took part in the study. CRN North London contacted 140 GP East London practices by email (Tower Hamlets, Newham and National Health Service City & Hackney Clinical Commissioning Groups (CCGs)), of which only two replied and participated in the study. Five of the eight GP practices contacted in North London (Barnet CCG) through a research coordinator took part in the study.

Patients with stroke and their caregivers were sent the postal questionnaire according to the following criteria.

### Inclusion criteria
#### Patients
► All patients aged >18 on the practice stroke register with documented history of stroke.

#### Caregivers
► Anyone identified by the patient as having a role helping with medicine taking.

### Exclusion criteria
► Patients who suffered a transient ischaemic attack (TIA) but not a stroke.
► Palliative or end-of-life patients.
► Patients receiving institutional long-term care (receiving total care in residential homes or living in nursing homes).
► Patients considered unsuitable to take part in the study by their GP.

### Survey participant identification
A list of prospective patients was compiled from the stroke register of each surgery by the practice staff. No

restriction was placed on the recruitment of survivors who were dependent for ADLs or lacking capacity. The list was screened by a practice GP and anyone not meeting the inclusion criteria or who was considered unsuitable for the study was excluded. Reasons for unsuitability were not collected for practical reasons.

## Survey participant recruitment

Eligible participants were sent a study survey pack by practice staff between September 2016 and February 2017. Study recruitment packs included two invitation letters, information sheets, questionnaires and postal version of Barthel Index,[28] one of which was for completion by the patient and the other by the caregiver. The Barthel Index provides a measure of functional independence and physical functioning and has been used in stroke research previously.[29] Patients with Barthel score of 20 were categorised as independent for ADLs, those with score 15–19 moderately dependent for ADLs and those with scores 0–14 severely dependent.[30] If receiving help with medications, the patient was asked to pass to their caregiver the invitation letter and information sheet and invite him/her to complete their copy of the questionnaire, providing answers on the patient's medicine taking. Family members, friends or paid caregivers of stroke survivors who were severely disabled and/or lacked mental capacity were invited to fill and return the caregivers' questionnaires only on behalf of patients. The information sheets stated that consent was implied by returning the completed questionnaire. Participants were asked to return completed questionnaires to the research centre in the Freepost envelopes provided. A second mail out of the study invitation pack was sent to all patients as a reminder, 2 weeks after the first one.

## SURVEY ANALYSIS

Survey data entry was performed by Document Capture Company.[31] Individual patients' characteristics (age, gender, time since stoke, number of daily medicines) were collected from the questionnaires themselves. Practice population, number of patients on stroke registers, deprivation score and ethnicity were taken from the National General Practice profiles (https://fingertips.phe.org.uk/profile/general-practice). The proportions of patients in each sociodemographic category, needing help taking medication, missing any medication in the previous 30 days and reporting the need for more help taking medication were estimated. When the survivor and caregiver questionnaires were both returned together, study data were collected from the patient's questionnaire only. The associations between 'unmet needs' and age (<or ≥70 years), gender, total number of medicines taken, dependence for ADLs, years since stroke and receiving help with medicines were investigated with individual logistic regression models (a different model per variable investigated), adjusted each and all of them for age and gender. Individual logistic regression models adjusted for

age and gender were also used to estimate the association between 'missed medicines in the previous 30 days' and age (< or >70 years), gender, total number of medicines taken, dependent for ADLs, years since stroke, help with medicines and unmet needs (a different model adjusted for age and gender per variable investigated). Sensitivity analysis was conducted to investigate if predictors of missing medication or unmet needs vary when the analysis was done on the whole dataset versus on questionnaires filled by patients only.

$\chi^2$ tests were used to compare the responses on unmet needs and missing medication given by patients versus caregivers.

All statistical analysis has been conducted with Stata V.14 (StataCorp).

## RESULTS

### Questionnaire development

Taking medications emerged as an important issue in all three workshops: nearly half of patients stated that a family member or friend was supporting them with daily medicine routines especially in relation to prompting medicine taking. This was put down to effects of the stroke itself on memory retention rather than general memory problems that people without stroke also experience. They admitted missing doses due to forgetting. Only a small proportion of survivors were actually handling their own prescriptions and were relying on support from family and/or community services. In one workshop, almost all survivors had dosette medication boxes and agreed that taking medications out of safety bottles and blister packs was a problem due to physical disabilities.

Thematic analysis of workshop data revealed five main practical domains of support needed with medication taking: (1) dealing with prescriptions and collection of medicines; (2) getting medicines out of the box, blister packs of bottles; (3) prompting 'It's time to take your medicine'; (4) swallowing medicines and (5) checking whether medicines have been taken. The final study questionnaire (see online supplementary file 1) included questions relating to each of these five domains, one item related to adherence (missed medicine in the last 30 days) and an assessment of disabilities through completion of the validated postal version of the Barthel Index.[28] The questionnaire was adapted for caregivers (see online supplementary file 2).

### Questionnaire finalisation

On the basis of the fourth workshop and two 'think-aloud' interviews, we reworded the survey questions (eg, from 'Do you get help with' was changed into 'Is somebody helping you with') and used a scale response 'all the time', 'often', 'sometimes', 'rarely', 'never' for the first question of each of the five survey domains, which was originally conceived as a 'yes' or 'no' answer (see online supplementary files 1,2 for text of questions).

## SURVEY

### Practice characteristics

Eighteen GP practices agreed to take part in the study, of which just over 1/3 were in London (n=7). GP practices were relatively large with an average population of 11 904 patients (SD=4010) and a low to moderate level of deprivation (Index of Multiple Deprivation[32]: mean 7.05: SD 3.19). Out of 3066 patients on the stroke registers, 1687 patients with stroke (55%) were considered eligible for the study and received the postal questionnaire. The average response rate of East of England and London practices was 42% and 27%, respectively. The response rate varied between 16% and 53% across practices.

### Participant characteristics

A total of 596 participants returned a completed questionnaire (549 (92.1%) from patients, 47 (7.9%) from caregivers showing a mean response rate of 35% (0.33–0.37)). Participants were on average 72.7 years old; 37.8% (n=210) of the sample were female (see table 1). There were a high proportion of white patients in the recruited practices (79%), which were on average 21% of mixed or ethnic minority background. Approximately 28% of study participants were completely independent for ADLs.

Participants getting any kind of help with medicines were on average 73.6 years old, two-thirds were male with only 19% of this group completely independent for ADLs.

Patients with unmet needs were on average 69 years old, predominantly male (71%) and 56.86% were severely dependent for ADLs. Patients who missed medications were on average 70 years old, 64% were male and the majority (48%) were moderately dependent for ADLs.

### Support with daily medication taking

Overall, 55.5% (95% CI 51.7 to 59.7) of the participants received help in at least one aspect of taking medication, in that they ticked one of the options from 'all the time' to 'rarely' on one or more of the five questions related to medicine taking. Eleven per cent (95% CI 8.8 to 13.9) of patients reported experiencing unmet needs and needing more help with at least one of the aspects of taking medication, in that they ticked 'yes' to the question 'do you feel you need more help' on one or more of the five questions related to medicine taking. The proportion of questionnaires reporting unmet needs filled in by caregivers, 19.6% (n=9), and by patients, 10.7% (n=57), had no significant difference (P=0.068).

Among the participants, help was needed to some degree with prescriptions and collection of medicines (49.8%), getting medicines out of the box or packet (27.9%), reminding to take medicines (36.4%), swallowing medicines (20.2%) and checking that medicines have been taken (34.1%) (see table 2). Being reminded to take medicines, dealing with prescriptions and collection of medicines and getting medicines out of a pack or bottle were the most commonly reported areas of unmet needs. Almost two-thirds of participants (65.3%) reported never missing medicines in the last 30 days. Out of the

**Table 1** Characteristics of participants who took part in the survey study (mean scores reported unless otherwise stated)

| | All patients | | | | Patients who receive any kind of help | | | | Patients with unmet needs | | | | Patients who miss medication | | | |
|---|---|---|---|---|---|---|---|---|---|---|---|---|---|---|---|---|
| | N | % | Mean | SD | N | % | Mean | SD | N | % | Mean | SD | N | % | Mean | SD |
| Age | 588 | | 72.7 | 11.6 | 331 | | 73.6 | 12.2 | 64 | | 68.8 | | 203 | | 70.5 | 13.0 |
| Female | 210 | 37.8 | | | 112 | 36.2 | | | 18 | 28.6 | | | 68 | 35.6 | | |
| Male | 346 | 67.2 | | | 197 | 63.2 | | | 45 | 71.4 | | | 123 | 64.4 | | |
| Time since stroke | 535 | | 7.7 | 7.6 | 295 | | 7.97 | 8.5 | 61 | | 9.3 | 9.2 | 186 | | 7.7 | 8.5 |
| N of daily medicines | 557 | | 6.4 | 4 | 312 | | 7.3 | 4.5 | 59 | | 9.7 | 7.1 | 190 | | 6.9 | 4.1 |
| Independent for ADLs (BI=20) | 139 | 28.3 | | | 53 | 18.9 | | | 5 | 9.8 | | | 45 | 25.7 | | |
| Moderately dependent for ADLs (BI=15–19) | 231 | 47.1 | | | 130 | 46.4 | | | 17 | 33.3 | | | 84 | 48.0 | | |
| Severely dependent for ADLs (BI=0–14) | 121 | 24.6 | | | 97 | 34.6 | | | 29 | 56.9 | | | 46 | 26.3 | | |

N represents the number of participants who completed the survey in respect to the different variables.
BI, Barthel Index.

**Table 2** Results summarising participants' responses to the survey questions

| | N | All the time N (%) | Often N (%) | Sometimes N (%) | Rarely N (%) | Never N (%) | Yes N (%) | No N (%) |
|---|---|---|---|---|---|---|---|---|
| Question 1<br>Is somebody helping with prescriptions and collection of your medicines? | 583 | 186 (31.9) | 19 (3.3) | 40 (6.9) | 45 (7.7) | 293 (50.2) | | |
| Question 1a<br>Do you feel you need more help with prescriptions and collection of your medicines? | 551 | | | | | | 33 (6.0) | 518 (94.0) |
| Question 2<br>Is somebody helping you getting the medicines out of the box, bottle or blister pack? | 578 | 85 (14.7) | 15 (2.6) | 31 (5.4) | 30 (5.2) | 417 (72.1) | | |
| Question 2a<br>Do you feel you need more help with getting the medicines out of the box, bottle or blister pack? | 553 | | | | | | 33 (6.0) | 520 (94.0) |
| Question 3<br>Is somebody helping with reminding you when is the time to take your medicine? | 577 | 78 (13.6) | 22 (3.8) | 59 (10.2) | 51 (8.8) | 367 (63.6) | | |
| Question 3a<br>Do you feel you need more help with reminding when is the time to take your medicine? | 564 | | | | | | 35 (6.2) | 529 (93.8) |
| Question 4<br>Is somebody helping you with swallowing your medicine? | 579 | 56 (9.7) | 11 (1.9) | 29 (5.0) | 21 (3.6) | 462 (79.8) | | |
| Question 4a<br>Do you feel you need more help with swallowing your medicine? | 560 | | | | | | 9 (1.6) | 551 (98.4) |
| Question 5<br>Is somebody helping you with checking that you have taken your medicines? | 576 | 76 (13.2) | 23 (4.0) | 58 (10.0) | 40 (6.9) | 379 (65.9) | | |
| Question 5a<br>Do you feel you need more help with checking that you have taken your medicine? | 558 | | | | | | 20 (3.6) | 538 (96.4) |
| Thinking of the last 30 days, how often did you miss taking your regular medicines? | 594 | 4 (0.7) | 5 (0.8) | 55 (9.3) | 142 (23.9) | 388 (65.3) | | |

**Table 3** Results of the multivariable analysis showing the variables associated with unmet needs

| Variable | Univariable analysis | | Multivariable analysis | |
| --- | --- | --- | --- | --- |
| | N | OR (95% CI) P value | N | OR (95% CI) P value |
| Age ≥70 | 581 | 0.6 (0.4 to 1.1) P=0.084 | 544 | 0.7 (0.4 to 1.2) P=0.180 |
| Gender (female) | 544 | 0.7 (0.4 to 1.2) P=0.137 | 544 | 0.7 (0.4 to 1.2) P=0.147 |
| Number of different medicines | 542 | 1.2 (1.1 to 1.3) P<0.001 | 509 | 1.2 (1.1 to 1.3) P<0.001 |
| Moderate dependence for ADLs (BI: 15–19) | 479 | 2.2 (0.8 to 6.1) P=0.135 | 447 | 2.7 (1.0 to 7.5) P=0.068 |
| Severe dependence for ADLs (BI: 0–14) | 479 | 8.5 (3.2 to 22.8) P<0.001 | 447 | 11.6 (4.2 to 32.4) P<0.001 |
| Years since stroke | 522 | 1.0 (1.0 to 1.1) P=0.078 | 490 | 1.0 (1.0 to 1.1) P=0.160 |
| Getting help with prescriptions and collection of medication | 568 | 4.7 (2.5 to 8.8) P<0.001 | 533 | 4.6 (2.4 to 8.7) P<0.001 |
| Getting help with taking medicines out of the box, bottle or blister pack | 563 | 6.7 (3.8 to 11.8) P<0.001 | 527 | 6.6 (3.6 to 11.8) P<0.001 |
| Getting help with reminding you when is the time to take your medicine? | 562 | 4.7 (2.7 to 8.2) P<0.001 | 526 | 4.3 (2.4 to 7.6) P<0.001 |
| Getting help to swallow the medication | 565 | 6.7 (3.9 to 11.6) P<0.001 | 528 | 6.8 (3.8 to 12.0) P<0.001 |
| Getting help by checking that you have taken your medicines | 562 | 4.9 (2.8 to 8.6) P<0.001 | 526 | 5.9 (3.1 to 10.1) P<0.001 |
| Getting any kind of help | 574 | 5.9 (2.8 to 12.1) P<0.001 | 537 | 5.9 (2.7 to 11.6) P<0.001 |

ADLs, activities of daily living; BI, Barthel Index.

34.7% of patients who said they missed taking medicine at any point in the previous 30 days, 23.9% said rarely, 9.3% sometimes, 0.8% often and 0.7% all the time. The proportion of questionnaires reporting missing medication at some point, filled in by caregivers, 27.7% (n=13), and by patients 35.3% (n=193), had no significant difference (P=0.292).

### Factors associated with unmet needs

Being on a higher total number of daily medications (OR: 1.2, (1.1 to 1.3), P<0.001), severe dependence for ADLs (OR: 11.6 (4.2 to 32.4) P<0.001) and receiving any kind of help (OR: 5.9, (2.7 to 11.6), P<0.001) in relation to taking medication was associated with experiencing unmet needs. Getting help with swallowing medicines (OR: 6.8, (3.8 to 12.0), P<0.001) and getting medicines out of a box, blister packs or bottles (OR: 6.6, (3.6 to 11.8), P<0.001) showed the strongest associations with experiencing unmet needs (see table 3).

When the analyses were conducted with data from questionnaires filled by patients only, the variables significantly associated with unmet needs were the same, apart from years since stroke (see online supplementary appendix 1).

### Factors associated with missing medications

Being older (age ≥70) was associated with a lower probability of missing medication (OR: 0.6 (0.4 to 0.9) P=0.006). Being on a higher number of daily medicines (polypharmacy) (OR: 1.1 (1.0 to 1.1), P=0.008) and getting any kind of help with medicine taking (OR: 2.1 (1.4 to 3.0) p<0.001) was associated with higher probability of missing medicines. The more unmet needs stroke

survivors had with taking medication, the more likely they were to miss their medicines (OR: 5.3 (3.0 to 9.4), P<0.001) (see table 4). When the analyses were conducted with data from questionnaires filled by patients only, the variables significantly associated with missing medication were the same (see online supplementary appendix 1).

### DISCUSSION
#### Summary of findings

From workshops, we identified five key issues that patients regarded as important with medication taking after stroke. We converted these into a five-item questionnaire that we distributed to people on stroke registers in 18 general practices. We obtained a response rate of 35%. Among respondents, 56% of survivors in the community were receiving help in some aspect of daily medication taking, 11% reported needing more help in at least one domain of medicine taking and 34% missed taking their medicines at some point in the previous 30 days.

A higher total number of daily medicines, being severely dependent for ADLs and receiving help with medication, were predictors of experiencing at least one unmet need in respect of medication taking. Stroke survivors who were younger, taking a higher number of daily medicines and experiencing a greater number of unmet needs were more likely to miss medications.

This work identified issues from a population that includes patients severely affected by stroke, who are often excluded from research.[17] Results presented here shed light on the effect of stroke-related impairments on practical domains and predictors of medicine taking,

**Table 4** Results of univariable and multivariable analysis showing associations with missing medicines

| Variable | Univariable analysis | | Multivariable analysis | |
| --- | --- | --- | --- | --- |
| | N | OR (95% CI) P value | N | OR (95% CI) P value |
| Age ≥70 | 594 | 0.6 (0.4 to 0.8) P=0.003 | 555 | 0.6 (0.4 to 0.9) P=0.006 |
| Gender (female) | 555 | 0.9 (0.6 to 1.2) P=0.401 | 555 | 0.9 (0.6 to 1.3) P=0.498 |
| Number of different medicines | 555 | 1.0 (1.0 to 1.1) P=0.040 | 520 | 1.1 (1.0 to 1.1) P=0.008 |
| Moderate dependence for ADLs (BI: 15–19) | 490 | 1.2 (0.8 to 1.8) P=0.468 | 456 | 1.3 (0.8 to 2.0) P=0.343 |
| Severe dependence for ADLs (BI: 0–14) | 490 | 1.3 (0.8 to 2.1) P=0.342 | 456 | 1.4 (0.8 to 2.4) P=0.239 |
| Years since stroke | 533 | 1.0 (0.9 to 1.0 P=0.950 | 499 | 1.0 (0.9 to 1.0) P=0.971 |
| Getting help with prescriptions and collection of medication | 581 | 2.0 (1.5 to 2.9) P<0.001 | 544 | 2.3 (1.6 to 3.3) P<0.001 |
| Getting help to have the medicines out of the box, bottle or blister pack | 576 | 1.4 (1.0 to 2.0) P=0.089 | 538 | 1.5 (1.0 to 2.2) P=0.051 |
| Getting help with reminding you when is the time to take your medicine | 575 | 2.5 (1.7 to 3.6) P<0.001 | 537 | 2.7 (1.8 to 3.9) P<0.001 |
| Getting help to swallow the medication | 578 | 1.5 (1.0 to 2.3) P=0.045 | 539 | 1.7 (1.1 to 2.6) P=0.022 |
| Getting help by checking that you have taken your medicines | 576 | 2.4 (1.7 to 3.4) P<0.001 | 537 | 2.5 (1.7 to 3.7) P<0.001 |
| Getting any kind of help | 587 | 2.1 (1.4 to 3.0) P<0.001 | 548 | 2.1 (1.4 to 3.0) P<0.001 |
| Unmet needs (participant reported more help needed) | 580 | 5.3 (3.0 to 9.2) P<0.000 | 544 | 5.3 (3.0 to 9.4) P<0.001 |

ADLs, activities of daily living; BI, Barthel Index; N, number of observations.

which are shown to have significant effects on overall adherence.

## Strengths and limitations

A strength of this study is that the questionnaire was developed from patients' and caregivers' own views gathered through workshops. Although not recruited through purposive sampling, workshop participants suffered from a range of stroke-related impairments, as highlighted by the reported use of dosette boxes, dependence on others for aspects of medicine taking like prompting medication times and dependence for ADLs such as collecting prescriptions and taking tablets out of boxes. In the postal survey, the inclusion of stroke survivors regardless of level of dependence for ADLs permitted investigating a population who are understudied,[17] yet may have significant unmet needs that can affect their adherence to medications. This investigation highlights caregivers' role in managing medicines in survivors dependent for ADLs.

However, study limitations should also be considered. The response rate across recruited GP practices was low and harder to reach stroke survivors may have been missed. Poor response rate is a source of bias that might affect our estimates.

Interestingly, considering the average age at stroke in England (ie, 74 for men and 80 years for women),[19] our participants' population was slightly younger (73 years), perhaps reflecting the fact that patients receiving institutional long-term care were excluded from the study or that older people found it harder taking part in a postal survey. Through the Barthel score, we did not assess cognition directly, although low cognitive function is associated with poor adherence.[33] As the Barthel focuses on physical disability, it is not known to what extent study participants were cognitively impaired or suffered from communication difficulties like aphasia. In addition, dependency for ADLs could have been caused by existing comorbidities other than stroke. We did not collect information on the use of blister packaged medication or devices to aid compliance, which could have influenced medication-taking practices. Finally, this study examined all medicine taking and did not differentiate between stroke secondary prevention medications and other drug categories.

## Comparisons with existing research

To our knowledge, this is the first study that shows that more than half of all stroke survivors get help with some aspect of medicine taking and that those receiving help are more likely to have unmet needs. This provides some insight in to why adherence to medication in stroke survivors may be poor.[34]

Moreover, the greater the number of medicines, the more likely stroke survivors were to miss medications. Addressing pill burden by simplifying drug regimens may be an important focus for future interventions. Indeed the polypill approach to medication taking has been shown to reduce cardiovascular as well as total pill burden in a primary care setting.[35] Simpler dosing regimens are known to be associated with better medication

adherence,[36] while fewer medicines has been shown to be an independent predictor of long-term medication persistence among stroke survivors.[37 38] A recent trial incorporating a fixed-dose combination polypill approach to taking cardiovascular medicine demonstrated better adherence among patients receiving a single pill.[39]

Receiving help with prescriptions and collecting medicines was identified as the area where most help was received (49.7% of respondents). Stroke survivors who are dependent for ADLs may face considerable practical challenges accessing healthcare resources at the pharmacy and the GP practice. A recent study in the USA found that around two-thirds of caregivers were involved in at least one medication management activity of elderly patients and that high involvement in instrumental ADLs was associated with the caregiver providing the patient with assistance in ordering medicines.[40] Filling prescriptions is also known to be an important factor influencing medication adherence.[41 42] Indeed caregivers can play a significant role in ensuring appropriate medication taking. A recent interview study exploring potential barriers and facilitators of medication adherence in stroke identified the central role of the caregiver in medication adherence.[43] Our evaluation of an online stroke forum also confirmed the important role of the caregiver in facilitating medication adherence.[14] Monitoring prescription collections, liaising with the GP and pharmacy, increasing the time between prescriptions or arranging medication deliveries, may help to address prescription needs.

Around 11% of stroke survivors reported unmet medication needs. We found that stroke survivors severely dependent for ADLs and receiving help with medicines were more likely to report unmet needs, which is in line with a recent study investigating stroke/TIA survivors in Australia, where greater functional ability was associated with fewer unmet needs, including those related to secondary prevention.[44] In previous research on unmet needs among stroke survivors, a 44-item survey study by McKevitt and colleagues reported that 49% of stroke survivors had at least one unmet need,[25] while in a study of Australian survivors who completed a 58-item survey, the percentage was 84%.[24] Both these studies however, examined unmet needs over a variety of domains including health, work, leisure and everyday living, social support and finances, whereas our study focused on medication needs only.

Getting help to take medicines out of a box, packet or bottle was the area where the greatest proportion of stroke survivors needed help all of the time. We previously found that the use of pill boxes and blister-packed medication to be both a facilitator[35] and a barrier[14] to adherence among stroke survivors,[15] while interventions using blister packaging and pill boxes have been found to be associated with improved adherence.[45] Although electronic medication devices were considered potentially effective in improving medication-taking behaviour among patients with cognitive impairments, success in using such devices was dependent on the patient having a good level of dexterity, while removing the medication from these devices was also found to be challenging.[46–48]

The need for further support in this domain, as reported in the current study, suggests that handling medications remains problematic for stroke survivors.

An interesting finding from this survey study is that stroke survivors who missed medicines were younger. This is consistent with other research on adherence in stroke that found that younger age was predictive of poor adherence[49] and has also been described in patients taking medication for cardiovascular disease.[50] The finding in the present study contrasts with the view that older patients are more likely to face difficulties taking medication,[51 52] which is frequently attributed to higher number of pre-existing comorbidities resulting in polypharmacy and increased complexity of medication-taking regimens. The fact that older patients may be less likely to miss medicines might be down to the support they receive from caregivers. Our findings suggests that support needed with medications may be overlooked in younger stroke survivors.[53]

In this study, a significant proportion of patients admitted missing medications occasionally. There is evidence that improving adherence by one antihypertensive pill/week for a once-a-day regimen reduces the hazard of stroke by 8%–9% and death by 7%.[54] Each incremental 25% increase in proportion of days covered with statin medications is associated with a 0.10 mmol/L reduction in low-density lipoprotein cholesterol.[55 56] Non-adherence to cardiovascular medications is associated with increased risk of morbidity and mortality.[57]

## Implications for clinical practice

A significant proportion of patients, particularly those who take large numbers of tablets, are disabled or receive help to take medication, have unmet needs and miss their tablets, which can increase risk of recurrent cardiovascular events. These particularly vulnerable groups of patients might benefit from focused clinical attention. Through understanding the needs of survivors and caregivers in different aspects of daily medication taking, we can help direct future resources to the areas of greatest need. For example, further exploration of medication packaging is warranted to understand the difficulties stroke survivors face handling medicines. Polypharmacy remains a difficulty for older patients. Therefore, exploring the use of combination pills and further efforts to reduce the burden of multiple medications among stroke survivors is warranted.

The questionnaire we have developed could be used to understand the challenges around medication faced by other patient groups. Unmet medication needs among UK stroke survivors have not been previously explored in the context of activities both survivors and caregivers consider important for taking medicines. Through understanding the extent of unmet needs as well as the areas in which these are greatest, strategies can be developed

which address poor medication-taking practices and therefore improve medication adherence.

## Future research

Novel interventions focussing on the practicalities of taking medicines and aimed at improving stroke survivors' adherence to treatment are needed. The findings reported here may inform the development of such interventions. Advances in technology have the potential to facilitate delivery of such interventions, for example, electronic devices prompting medication-taking times.[58 59] Efforts to improve medication taking among survivors of stroke using technology are already under way and have shown promise.[60]

**Acknowledgements** The authors wish to thank all the stroke survivors and caregivers who participated in this study.

**Contributors** ADS is the chief investigator, contributed to the study design, data analysis and commented on the manuscript. JJ contributed to the study design, data collection, data analysis and prepared the manuscript for submission. JM and SS are coinvestigators on the study, wrote and commented on the manuscript. LA and GLDT contributed to the data analysis and commented on the manuscript. All authors agreed on the final draft of the submitted manuscript.

**Funding** This study was funded by the RCGP SFB, Ref. SFB 2014–15 'Quantifying the support stroke survivors get with daily medication taking: a questionnaire survey'. ADS and LA are funded by NIHR Academic Clinical Lectureships. This article therefore presents independent research funded by NIHR. JJ was supported by a research grant from The Stroke Association and the British Heart Foundation: TSA BHF 2011/01.

**Disclaimer** The views expressed are those of the authors and not necessarily those of the NHS, the NIHR or the Department of Health.

**Competing interests** None declared.

**Patient consent** Obtained.

**Ethics approval** This study has received ethical approval from Cambridge Central Research Ethics Committee (REC reference: 16/EE/0182) and from the Health Research Authority (IRAS project ID: 170931).

**Provenance and peer review** Not commissioned; externally peer reviewed.

**Data sharing statement** No other data available.

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
