## [Reviewer comments · BMJ Open]

ARTICLE DETAILS

TITLE (PROVISIONAL)	Evaluating practical support stroke survivors get with medicines and unmet needs in primary care: A survey
AUTHORS	JAMISON, JAMES Ayerbe, Luis Di Tanna, Gian Luca Sutton, Stephen Mant, Jonathan De Simoni, Anna

VERSION 1 – REVIEW

REVIEWER	Esteban González-López Family Medicine and Primary Care Division. School of Medicine. Autonoma University of Madrid, Spain
REVIEW RETURNED	16-Oct-2017

GENERAL COMMENTS	The authors pointed that the main limitation was the lower rate of responses. May be some results would have been different if the responses' rate were higher. Barthel's test is a good way to measure disability and ADL, but I would like to suggest the authors to include in their future research other tests such as Lawton-Brody, JH Downtown and Morisky-Green in order to check the adherence to drugs regime.
---

REVIEWER	Sureshkumar Kamalakannan Public Health Foundation of India India
REVIEW RETURNED	20-Oct-2017

GENERAL COMMENTS	The need for studies like this is very important and it is the need of the hour. However, I feel the entire manuscript lacks clarity especially on the conceptualization of the study, its methods and the relationship between the results and the its implications. There needs to be substantial explanation and justifications about the results and its implications. The author also provided a marked copy with additional comments. Please contact the publisher for full details.
---

REVIEWER	Professor Amanda Thrift Monash University, Australia
REVIEW RETURNED	30-Oct-2017

GENERAL COMMENTS	The authors outline development of a survey to assess the needs of survivors of stroke in regard to taking medications. They then determined the factors associated with unmet needs, and missing medications, in a group of patients with stroke and caregivers within a general practice sample. The methods are sound, although the manuscript could benefit from some reduction in duplication. The following may improve the value of the manuscript:  1. The Barthel Index could be better separated into categories (Table 1) to give a sense of the extent of disability. In particular, it would be of interest to know whether there is a dose-response relationship between the extent of disability and unmet needs. Please re-do analyses using commonly defined categorisations. 2. One of the limitations of the study is in the use of the Barthel index as a measure of disability. This scale has considerable floor and ceiling effects, and this limitation should be noted. 3. Table 1 would be simpler to compare if the columns and rows were transposed. 4. The number of participants do not always tally. For example, the authors state that they received completed questionnaires from 549 patients and 45 caregivers. This adds to 594 completed questionnaires, and not the 596 reported. Please check and correct all figures. 5. There are numerous areas where the wording could be reduced. For example, when stating that 37.8% were female, there is no need to state that the remaining proportion (62.2%) were male. Further reductions in the results could be reducing duplication between the text and the data in the tables. I'd suggest and highlighting just the major points in the text. 6. The absence of a significant association between age (sex and number of years since stroke) and unmet needs does not mean that there is no association. In reality, it may simply be an inability to detect a difference. This is particularly so for number of years since stroke and unmet needs. Please amend. 7. It is unclear how many people are included in the analyses in Tables 3 and 4. From Table 2 it is clear that there are some missing responses to each question. Please include the number of observations for each analysis in the legend. 8. What criteria were used to determine which variables should remain in the multivariable analyses? There does not appear to be justification for including 'years since stroke' in the multivariable analysis of missing medications. 9. The titles of the tables could be more explicit so that these can be understood without reference to the text. 10. It would be interesting to note the caregivers' perceptions of unmet needs (or missing medications) in comparison to the patients themselves. For example, it may be that caregivers who are providing medications to patients may not recognize this as an unmet need. It would also be worthwhile conducting a sensitivity analysis excluding responses from caregivers to determine whether there are any differences in interpretation of the results.
--

REVIEWER	Rosie Shannon Bradford Institute for Health Research, UK
REVIEW RETURNED	31-Oct-2017

GENERAL COMMENTS	• An enjoyable study surveying medication taking for stroke
---

survivors – study findings highlight the unmet needs, dependence on receiving help, and missed medications of an especially vulnerable group. This is an important step in order that these issues can be addressed.

ABSTRACT

- Conclusion page 3 line 5 - reads as though it is 1 in 10 of the half. Is this what is meant? Perhaps: 'More than half of patients who replied needed help with taking medication and, and 1 in 10 reported unmet needs regarding taking medication' ?
- Page 3 line 22 – Is this a list of Strengths and limitations or an article summary? If the former, the first two bullets points are a repetition of the study description/abstract, and could be replaced with points from Discussion strengths and limitations. If the latter, this is a suitable summary.
- Page 3 line 33 - 'Results shed light' could be a separate bullet

INTRODUCTION

- Page 4 line 40 – should this be 'handling'?
- End of sentence on page 5 line 14/15 is a repetition of line 4/5 – re proportion patients relying on caregivers

METHODS

- While the postal survey provides the main study findings, questionnaire development would appear crucial to this being a high quality survey, ideally would have liked to know more about this, this would add to overall survey robustness. Eg:
 - o All of the survey development arose purely from workshops. Was there any literature review or prior ideas which contributed topics which the team anticipated were of importance to include in the questionnaire?
 - o "recruitment was opportunistic" – how and where were participants at the first three workshops found, approached, and recruited, was consent was obtained, etc?
 - o Were workshops audio recorded? Did field notes provide sufficient data?
 - o How were workshops structured? eg topic guide/prompts?
 - o Are workshop participant characteristics of interest? – eg age, time since stroke, living alone or not, formal home care or not, ethnicity, cognitive impairment or aphasia.
- Inclusion criteria – caregivers: how was it known when a caregiver was identified (eg at the GP/coding stage and then letters targeted/addressed to them as such, or were carers simply identified if the questionnaire was picked up at the patients home by someone who identified themselves as a carer?)
- Page 6 line 50 sentence starting – "No restriction..." – I'm unsure if this makes sense – perhaps remove the word 'experiencing'?
- Page 6 line 5 – 'Terminally ill' would fall under exclusion criteria already listed ('palliative or end of life'), so what is different about exclusions made by GPs? It is important to clarify why people were excluded. Perhaps they were at the discretion of GP and their reasons may not be known, or GPs had their own criteria, if so this could be added to the list of exclusion criteria (eg 'GP considered them unsuitable'). A flowchart would be helpful.
- Page 7 line 5 – Did study packs included two of everything (one for carer and one for patient)? (if so could make this clearer by removing 'patient' from 'patient information sheets')
- Would detail on the workshops be of interest? findings, analysis, quotes. Why stopped at three workshops? How long did workshops last?
- Page 7 line 15 – "invite him/her to complete their copy" – is this the carer copy? Was the method that patients who receive any help were only expected to return a carer questionnaire an not a patient

	questionnaire?  • Page 7 line 55 "...data were collected from the patient's questionnaire only" This reads as though both patient and caregiver invited to respond, but only one of these data used. RESULTS  • Participant characteristics – " a high proportion of white participants" – provide percentage. • It could have been useful to know whether respondents live alone or not, have formal home care or not, have cognitive impairment or aphasia if available. • Table 1 – could add two rows, as well as those who receive help, also those with unmet needs, and those who missed meds – give that these three are the main arms of study findings. • Were all analyses conducted on the full sample of returned questionnaires? Was this n=596? • Page 10 line 12 – why is the 55.7% not 56.3% ie 331/588 as per table 1 ? • Table 2 – why is 'n' missing for 3a, 4a, 5a ? • Low return rate indicates poor acceptability of the survey, perhaps respondents felt the questionnaire didn't address their issues – is it possible to comment on or learn from any data about acceptability of the survey (eg in early testing, additional comments respondents may have written, or patterns in missing data)? • If needed to reduce word count maybe ORs, CIs and p values could be removed from the text as these are in the tables and the text adequately illustrates the main points. • An OR of 1.02 is reported as showing association (page 11 line 5), while a similar OR of 1.07 is reported as no association (page 11 line 19) DISCUSSION  • Page 12 line 7 – re taking a higher number of daily medications – keeping this depends on what you decide re difference between OR of 1.07 & 1.02. • Page 12 line 11 & line 34 (also page 3 line 31), data is provided on those who are independent according to Barthel, but no data on those more dependent, making it difficult to be sure whether this study includes patients "severely affected by stroke". One option could be including more dependent Barthel scorers in another column in table 1. A study method which is inherently biased against include those with cognitive impairment/aphasia (ie requires talking in a workshop or reading and writing the questionnaire), would need to be explicit about efforts to include them for this to be a representative stroke sample. If no efforts were taken this remains a useful and important survey of those who responded, but would be careful about wording around the diversity of this group unless demonstrated. • Page 12 line 31 – remove 'actual' • Page 15 line 31 – insert full stop after 'other patient groups'
--	--

VERSION 1 – AUTHOR RESPONSE

Editorial requirements:

- Please revise the 'Strengths and limitations' section of your manuscript. This section should relate specifically to the methods, and should not include a general summary of, or the results of, the study.
Response: the 'Strengths and limitations' section has been reviewed as suggested.

Reviewer(s)' Comments to Author:

Reviewer: 1

Reviewer Name: Esteban González-López

Institution and Country: Family Medicine and Primary Care Division. School of Medicine. Autonoma University of Madrid, Spain Please state any competing interests or state 'None declared': None declared

Please leave your comments for the authors below The authors pointed that the main limitation was the lower rate of responses. May be some results would have been different if the responses' rate were higher.

Barthel's test is a good way to measure disability and ADL, but I would like to suggest the authors to include in their future research other tests such as Lawton-Brody, JH Downtown and Morisky-Green in order to check the adherence to drugs regime.

Response: The authors recognise low response rate is problematic and have acknowledged this in the limitation section of the discussion.

We thank the reviewer for the suggestion on including an alternative measurement instrument for future research in this area.

Reviewer: 2

Reviewer Name: Sureshkumar Kamalakannan

Institution and Country: Public Health Foundation of India , India Please state any competing interests or state 'None declared': None

Please leave your comments for the authors below The need for studies like this is very important and it is the need of the hour. However, I feel the entire manuscript lacks clarity especially on the conceptualization of the study, its methods and the relationship between the results and the its implications. There needs to be substantial explanation and justifications about the results and its implications.

- Front page: this is cerebro vascular medicine or Cardio Vascular?

Response: This is Cardiovascular medicine. Cerebrovascular medicine was not available as a primary heading for the Journal.

- Page 1, line 3: Can be a bit reader friendly. Lacks clarity at present. Are you interested in Medication Adherence, unmet needs for support, independence in ADL. intended for stroke survivors or for primary care professionals. Could include where exactly.

Response: The aim was not to explore medication adherence among stroke survivors, but rather to explore practical areas of medication taking where stroke survivors. We have reworded the title into: Evaluating practical support stroke survivors get with medicines and unmet needs in primary care: A survey'

- Page 2, line 5: Title looked as if this was results from a survey. Development of questionnaire is mentioned primarily as a objective here.

Could also split the objectives as primary and secondary.

1. Design and development of a tool
2. Explore the unmet needs for Drugs/Medicines
3. Predictors for compliance and acceptance of medicines
4. Support for medication.

Response: The primary outcomes of this study are listed in the section 'Primary Outcome measures'.

Developing the questionnaire was undertaken as part of PPI exercises in the context of grant applications, and is now described in the methods, see page 5, lines 24-26. Workshops results have been already reported in a separate study (see reference 17).

- Page 2 , line 13: Is this a delphi process? Focus group Discussion, what is the composition of the workshop?

Response: Details of the workshops and numbers are provided on page 5, lines 21-27.

- Page 4, line 12: sentences can be rearranged. the last one going first and the first one going last

Response: The authors thank the reviewer for this suggestion and have now changed the paragraph with the last sentence now coming first. See page 4. Line 6-8.

- Page 4, line 24: this is contradictory to the first statement. which impacts on what?

Response: The first statement reads that being able to handle their own medication regimen may influence whether patients can manage their medication in their own home. Line 11-12 further confirms the point made, in that being disabled or dependent on others can negatively affect adherence to medication.

- Page 4, line 49: NO where up until now the age group related to stroke is mentioned. It will be great to have that statement before assuming that stroke survivors are all elderly ones.

Response: We thank the reviewer for this observation and have added a sentence describing the mean age at stroke in men and women in England, see page 4, line 25.

- Page 5, line 5. There must be clarity on our objective of whether you are looking at medication adherence for the elderly or for the stroke survivors. they are two different group of participants

Response: Our focus on patients with stroke (rather than the elderly) is highlighted in the title, objective section of the abstract and 'aims' section at the end of the Introduction. The focus of the study was not medication adherence, but finding out about practicalities of medicine taking after stroke, areas of unmet needs and predictors of unmet needs and missing medications. The target population included patient severely affected by stroke, who have been understudied so far.

- Page 5 line 11: the terminology could be better - did you mean medication adherence? Medication taking does not sound right.

Response: We meant medication taking. We have re-worded the title to reflect our focus on finding out about practicalities of medicine taking after stroke.

- Page 5, line 11: What kind of disabilities, physical, intellectual/cognitive, psychological

Response: This refers to all disabilities, whether physical or cognitive. This has now been clarified in the manuscript on page 5, line 7.

- Page 5, line 17: What are these aspects – please explain in detail.

Response: The authors have added a line to provide more detail on this, page 5, line 11.

- Page 5, line 21: support could be a better word compared to help in this instance.

Response: Thank you. The authors have changed the wording to read 'the support', as suggested. See page 5 , line 12. The abstract was also modified, see page 2, line 15.

- Page 5, line 35: There is no details related to the sample size estimation and the rationale.

Response: Thanks for pointing this out. Details regarding the sample estimation have now be added, see page 6, lines 7-11.

- Page 5, line 52: are the participants for the 4th workshop the same who attended the first three? and If everybody is from different strokes - There is question related to the replicability of the tool and representative sample from the population.

Response: Participants were different in all workshops, except from 1 participant of the second workshop who was also present at the first. We believe the fact that the same issues around practicalities and support needed with medications emerged from all workshops represent a strength in the process of development of the survey.

- Page 6 ,line 7: there was no mention about why this method compared to direct patient survey

Response: The postal version of the Barthel index has been validated, see reference Gompertz P, Pound P, Ebrahim S. A postal version of the Barthel Index. Clinical Rehabilitation 1994;8(3):233-39. doi: doi:10.1177/026921559400800308

This has been mentioned at page 9, line 16.

Postal surveys have been widely used to gather data from the stroke population. Our results and conclusions are framed in the context of a postal survey.

- Page 6,line 8 : why did you recruit participants through CRN and GP practices. why not from the stroke register or from community registers - this has a link to the numbers recruited. There is too many drop outs and low response rates.

Response: Our aim was to evaluate practical support stroke survivors get with medicines and unmet needs in primary care. Therefore our recruitment was setup to take place in GP practices. The NIHR Clinical Research Networks (in this study CRN Eastern England and CRN North Thames) help to increase the opportunities for patients to take part in clinical research, ensures that studies are carried out efficiently, and supports the Government's Strategy for UK Life Sciences by improving the environment for commercial contract clinical research in the NHS. Our study was adopted as NIHR portfolio study and therefore benefited of CRN support with GP practices recruitment. The authors acknowledge the low response rate as limitation of the research.

- Page 6, line 26: The introduction looked like the study was related to older adults only.

Response: We thank the reviewer for this observation that has given us the opportunity to add a line reporting that age at stroke was not a factor for patients experiencing difficulties with medications. See page 4, line 21-22.

- Page 6, line 31 : This is very crucial point. Support in many parts of the UK are provided through paid carers too. hence how do we differentiate the carer you mention here (sounding more like a family member) compared to the paid carer who has the responsibility to provide support for medication too. what are the carers characteristics in the study?

Response: When the survivor and caregiver questionnaires were both returned together, study data were collected from the patient's questionnaire only. Therefore out of 596 analysed surveys, 549 were

filled by patients and 47 by caregivers. Caregivers were asked about their relation to patients (see first page in the caregiver survey, in the manuscript supplements). Most of them were spouses, though patients' children and 'carers from an agency' were also represented. Because of the low number (47 out of 596), we did not include this in the manuscript.

- Page 6, line 35 : what about stroke participants with severe psychiatric or cognitive disorder. what about age-related comorbidities like dementia pre-stroke.

Response: There was no restriction on recruiting stroke patients with any level of disability. This meant that we could not distinguish between difficulties with medication taking due to specific stroke related impairments or to other pre-existing co-morbidities. We have added a line in the Discussion, see page 14, lines 1-3.

The information leaflets included in the study pack indicated that it was appropriate in severe psychiatric or cognitive disorder (i.e. patient lacking capacity) that the caregiver survey only was filled and returned, provided the caregiver felt that taking part in the survey reflected their family member's, friend's or patient's with stroke wishes, if he/she was able to competently express them. This was reviewed and approved by a special NHS Research Ethic Committee, with expertise in recruiting to research patients lacking capacity.

- Page 6, line 43 : please explain why this group is excluded. This group of participants are the apt ones for a study of this kind.

Response: We did not include patients receiving institutional long term care as most likely their medicine taking is under complete control of caregivers.

As stated in the Introduction, remaining independent at home may depend on how well patients can manage complex medication regimens. Our ultimate aim is to help patients remain independent in the community, by identifying unmet needs with medicine taking and putting in place appropriate primary care interventions to respond to such needs.

- Page 7, line 21 : Not sure you will use this data and justify the fidelity of this set of data. kindly add more details

Response: See our response two points above.

A sensitivity analysis has now been added to investigate if predictors of missing medication or unmet needs varied when the analysis was done on the whole dataset versus on questionnaires filled by patients only. This showed no significant difference between groups. See appendix 1.

- Methods, page 7, line 57: why was this done - The data completed by the carers - what was done with it? Please mention these areas in detail.

Response: Our responses to the points raised above shall answer this point, too.

- Page 8, line 3: Make it simple to understand. Proportion of people with unmet need, issues with medication adherence, etc.

Response: The paragraph describing the analysis of the survey has now been reworded in the manuscript, see page 8, lines 13-23.

- Page 8, line 25: The paragraph below reflects on my question related to inclusion criteria. Stroke has cognitive effects - the inclusion criteria should be clearly stating the characteristics of the participants for inclusion. including how it was assessed.

Response: Recruitment of participants to workshops was opportunistic in the context of PPI exercises. Although purposive sampling was not applied, participants included patients living alone and independent with medicine taking as well as patients who were dependent on caregivers (nearly half of participants, i.e. 12/26). As outlined also to a reviewer's 4 comment further down, during the 3 workshops to develop the questionnaire, the 12 caregivers were doing most of the talking on behalf of the patients they were looking after, who were moderately/severely disabled.

- Page 8, line 45: details about the questionnaire could be helpful. Construct, content, number of items, scoring etc.

Response: The questionnaires, including details of the questions, number of items and content, are shown in Supplementary File 1 and 2.

- Page 9, line 3: There needs to be substantial explanation about how finalisation was done. Was there any experts to help the investigators? How the decision on the method to questionnaire development came through, how did the investigators ensure that what the participants saying was correct. These should be explained.

Response: In the manuscript, page 9, lines 19-22 described questionnaire finalisation. Feedback from the 4th workshop where participants attempted themselves filling the questionnaires providing feedback onto each question, as well as two 'think aloud' interviews with stroke survivors were used to inform questionnaire finalisation, with survey questions being re-worded to reflect suggestions. e.g. questions were changed from 'Do you get help with' into 'Is somebody helping you with' and used a scale response 'All the time', 'Often', 'Sometimes', 'Rarely', 'Never' for the first question of each of the five survey domains, which was originally conceived as a 'yes' or 'no' answer. Questionnaire finalisation was also informed by the expertise of the study team consisting of an academic GP and experienced qualitative researchers.

- Page 9, line 21 : why only these practices, what is the rationale for selection of these practices. This must be explained.

Response: As reported on page 6, line 12, GP practices were approached by the CRNs. The CRNs were identified at the study funding application stage because of convenience, considering the research team being based in Cambridge (hence CRN Eastern England) and London (hence CRN North Thames).

- Page 9, line 31: there should also be a very clear explanation related to the response rates.

Response: The authors acknowledge that the low response rate is a limitation of this survey study. Response rate varied between top 53% in top and 17% in the poorest recruiting practices. No practice from Eastern England recruited less than 20% of patients, while 3 London practices did so. Nevertheless, other London practices recruitment was close to the average 35%, with the top recruiting practice showing a response rate of 40%. Details of response rates across practices are included in the manuscript, see page 10, lines 12-13.

We have added a paragraph mentioning a possible reason of our low response rate, i.e. older age and consequent difficulties with filling surveys. In fact, our participants' population is younger compared with the average age at stroke in England. See page 13, lines 25-28.

- Page 9, line 35 : how did you ensure that the characteristics were true. Was any records cross checked.

Response: Participant characteristics were provided individually through the filled surveys. A copy of the surveys to show what participants' characteristics were collected are included as supplementary files. The study protocol as approved by the NHS Research Ethic Committee did not include cross check of participants' characteristics with electronic patients' records.

- Page 9, line 47 : according to Barthel?

Response: we opted to use the widely accepted phrase 'dependence for ADLs' rather than referring to the actual Barthel score throughout the paper.

- Page 9, line 51: if the participants were selected from registers. recent strokes would have been easily identified. 8 years is too much time to understand whether old age is the problem or stroke itself is the problem.

Response: the observation is true, though even patients with a recent stroke might have pre-existing conditions (e.g. severe mental illness) that might have affected medication taking independently from stroke.

We have added to the study limitations that dependency for ADLs could have been due to existing co-morbidities other than stroke, see page 14, lines 1-3.

Stroke survivors continue to improve over a long time, sometimes over a number of years. Recovery of function can affect their abilities to independently take medicines.

As mentioned in one of our answers above, our ultimate aim is to help patients remain independent in the community, by identifying unmet needs with medicine taking and putting in place appropriate interventions to respond to such needs.

- Page 10, line 10: the numbers should be transparently reported.

Response: The numbers of observations for each analysis is now reported in the Tables.

- Page 12, line 22: the rationale for the limitations have to be discussed.

Response: we have provided more rationale around the study limitations, see page 13, lines 22-28 and page 14 line 1-6.

Reviewer: 3

Reviewer Name: Professor Amanda Thrift

Institution and Country: Monash University, Australia. Please state any competing interests or state 'None declared': None declared.

Please leave your comments for the authors below. The authors outline development of a survey to assess the needs of survivors of stroke in regard to taking medications. They then determined the factors associated with unmet needs, and missing medications, in a group of patients with stroke and caregivers within a general practice sample. The methods are sound, although the manuscript could benefit from some reduction in duplication. The following may improve the value of the manuscript:

1. The Barthel Index could be better separated into categories (Table 1) to give a sense of the extent of disability. In particular, it would be of interest to know whether there is a dose-response relationship between the extent of disability and unmet needs. Please re-do analyses using commonly defined categorisations.

Response: We have categorised the survey results according to the Barthel Index into three levels of disability commonly used in Stroke literature, i.e. Independent for ADL's (BI=20), Moderately

dependent for ADLs (BI=15-19), Severely dependent for ADLs (BI=0-14), see Table 1. The analysis has been re-done with this new categorization and tables 3 and 4 updated accordingly.

2. One of the limitations of the study is in the use of the Barthel index as a measure of disability. This scale has considerable floor and ceiling effects, and this limitation should be noted.

Response: The authors have now noted this limitation of the Barthel index in the Strengths and Limitations section of the manuscript on page 14 line 1-3.

3. Table 1 would be simpler to compare if the columns and rows were transposed.

Response: The authors thank the reviewer for this suggestion. In Table 1 the columns and rows have now been transposed to make it easier to read.

4. The number of participants do not always tally. For example, the authors state that they received completed questionnaires from 549 patients and 45 caregivers. This adds to 594 completed questionnaires, and not the 596 reported. Please check and correct all figures.

Response: The authors thank the reviewer for this observation. Numbers of participants have now been checked and corrected, i.e. there were 47 rather than 45 surveys returned by caregivers.

5. There are numerous areas where the wording could be reduced. For example, when stating that 37.8% were female, there is no need to state that the remaining proportion (62.2%) were male.

Further reductions in the results could be reducing duplication between the text and the data in the tables. I'd suggest and highlighting just the major points in the text.

Response: We have now removed unnecessary wording from the Results section to avoid duplication with what has been reported in the tables. In the results section- Participant characteristics- we removed the sentence detailing number of years since stroke and medicines being taken per day. In the Support with daily medication taking section, we removed the first sentence which describes what is displayed in Table 2. In the Factors associated with unmet needs section, we removed the sentence which reports no association between variables and unmet needs. In the Factors associated with missing medications section we also removed the sentence that reported no association between variables and missing medications.

6. The absence of a significant association between age (sex and number of years since stroke) and unmet needs does not mean that there is no association. In reality, it may simply be an inability to detect a difference. This is particularly so for number of years since stroke and unmet needs. Please amend.

Response: Thank you for this observation. As described in our response above, we have removed from the results the sentences about the lack of association between age (sex and number of years since stroke) and unmet needs. We have added the following sentence in the limitation section of the Discussion: Poor response rate is a source of bias that might affect our estimates, see page 13 line 23-24.

7. It is unclear how many people are included in the analyses in Tables 3 and 4. From Table 2 it is clear that there are some missing responses to each question. Please include the number of observations for each analysis in the legend.

Response: The number of observations included in the univariable and multivariable analyses has now been included in Table 3 and Table 4.

8. What criteria were used to determine which variables should remain in the multivariable analyses? There does not appear to be justification for including 'years since stroke' in the multivariable analysis of missing medications.

Response: Regression analysis included variables that could be of significance when studying associations with experiencing unmet needs and missing medicines.

Stroke survivors continue to improve over a long time, sometimes over a number of years. Recovery of function can affect their abilities to independently take medicines. We therefore have included time since stroke in the multivariable analyses.

9. The titles of the tables could be more explicit so that these can be understood without reference to the text.

Response: The authors have amended the title of each of the tables so that they can be more easily understood.

10. It would be interesting to note the caregivers' perceptions of unmet needs (or missing medications) in comparison to the patients themselves. For example, it may be that caregivers who are providing medications to patients may not recognize this as an unmet need. It would also be worthwhile conducting a sensitivity analysis excluding responses from caregivers to determine whether there are any differences in interpretation of the results.

Response: we have now added the sensitivity analyses suggested, comparing unmet needs and missing medications analysing responses separately, see page 11 lines 8-9, and lines 17-18. In the same way, we have additionally compared predictors of unmet needs and missing medication. See Appendix 1.

No significant differences were identified, with the exception of 'years since stroke' being significantly associated with unmet needs. This is now reported in the results section. The procedure used for these tests have been added in the methods, see page 8, lines 13-23.

Reviewer: 4

Reviewer Name: Rosie Shannon

Institution and Country: Bradford Institute for Health Research, UK Please state any competing interests or state 'None declared': None declared

Please leave your comments for the authors below

- An enjoyable study surveying medication taking for stroke survivors – study findings highlight the unmet needs, dependence on receiving help, and missed medications of an especially vulnerable group. This is an important step in order that these issues can be addressed.

ABSTRACT

- Conclusion page 3 line 5 - reads as though it is 1 in 10 of the half. Is this what is meant? Perhaps: 'More than half of patients who replied needed help with taking medication and, and 1 in 10 reported unmet needs regarding taking medication?'

Response: Thank you for this observation. What is meant is 1 in 10 of all participants. This has now been corrected, see page 3, line 2.

- Page 3 line 22 – Is this a list of Strengths and limitations or an article summary? If the former, the first two bullet points are a repetition of the study description/abstract, and could be replaced with points from Discussion strengths and limitations. If the latter, this is a suitable summary.

Response: This is a list of the Strengths and Limitations of the study. The authors acknowledge that the first 2 bullet points in this section are repetitive and have now reworded both points to better reflect the study strengths. See page 3, lines 12-14

- Page 3 line 33 - 'Results shed light' could be a separate bullet

Response: We agree with this observation and have made this sentence as an additional bullet point. See page 2, lines 17-19

INTRODUCTION

- Page 4 line 40 – should this be 'handling?'

Response: Thank you for this observation. The authors agree this word should read 'handling' and have now corrected this on page 4, line 20.

- End of sentence on page 5 line 14/15 is a repetition of line 4/5 – re proportion patients relying on caregivers

Response: Thank you for this observation. We removed the sentence on page 5, line 3-4 of the manuscript.

METHODS

- While the postal survey provides the main study findings, questionnaire development would appear crucial to this being a high quality survey, ideally would have liked to know more about this, this would add to overall survey robustness. Eg:

All of the survey development arose purely from workshops. Was there any literature review or prior ideas which contributed topics which the team anticipated were of importance to include in the questionnaire?

Response: An evaluation of current evidence was performed and previously published separately (see reference 17). We have added a line to the methods section, page 5, line 21.

- “recruitment was opportunistic” – how and where were participants at the first three workshops found, approached, and recruited, was consent was obtained, etc?

Response: the workshops were organised in the context of gathering Patient and Public Involvement (PPI) input into research grant applications aimed at improving adherence to medication after stroke (as described in reference 17). Page 5, line 24-26.

- Were workshops audio recorded? Did field notes provide sufficient data?

Response: Workshops were not audio recorded, however they were attended by a few researchers. Field notes provided sufficient data for workshop analysis.

- How were workshops structured? E.g. topic guide/prompts?

Response: The workshops discussions were structured using an agenda and a topic guide which consisted of a list of questions informed by the literature, expertise of the research team and fields notes of previous workshops (from the second workshop onward).

- Are workshop participant characteristics of interest? – eg age, time since stroke, living alone or not, formal home care or not, ethnicity, cognitive impairment or aphasia.

Response: Details of workshop participants including cognitive impairment, ethnicity and formal home care were not collected. Although purposive sampling was not applied, participants included patients living alone and independent with medicine taking as well as patients who were dependent on caregivers (nearly half of participants, i.e. 12/26).

- Inclusion criteria – caregivers: how was it known when a caregiver was identified (eg at the GP/coding stage and then letters targeted/addressed to them as such, or were carers simply identified if the questionnaire was picked up at the patients home by someone who identified themselves as a carer?)

- Response: A separate information sheet and questionnaire for caregivers were contained in all envelopes sent to the stroke survivors. Stroke survivors who were receiving any help with medications were invited to pass the information sheet and questionnaire to the caregiver. Caregivers were therefore identified by the stroke survivor themselves. Detail on this is included in the Methods section on page 7, lines 12- 15 and 18-22.

- Page 6 line 50 sentence starting – “No restriction...” – I’m unsure if this makes sense – perhaps remove the word ‘experiencing’?

Response: Thank you for this observation. The authors have now removed the word ‘experiencing’ which does not make sense in the sentence. See page 7, line 7.

- Page 6 line 5 – ‘Terminally ill’ would fall under exclusion criteria already listed (‘palliative or end of life’), so what is different about exclusions made by GPs? It is important to clarify why people were excluded. Perhaps they were at the discretion of GP and their reasons may not be known, or GPs had their own criteria, if so this could be added to the list of exclusion criteria (eg ‘GP considered them unsuitable’). A flowchart would be helpful.

Response: Thank you for noting this. We have added the exclusion criteria 'Patients considered unsuitable to taking part in the study by their GP' (see page 7, line 3). We have also removed the text between brackets '(e.g. terminally ill)', see page 7, line 9.

• Page 7 line 5 – Did study packs included two of everything (one for carer and one for patient)? (if so could make this clearer by removing 'patient' from 'patient information sheets')

Response: Yes. The study information pack included two of everything, one for the stroke survivor and one for the caregiver. The authors have removed the word 'patient' as suggested, see page 7 line 13.

• Would detail on the workshops be of interest? findings, analysis, quotes. Why stopped at three workshops? How long did workshops last?

Response: All workshops lasted approximately 1 hour. Three workshops were considered sufficient as this was the point at which no new data emerged.

• Page 7 line 15 – "invite him/her to complete their copy" – is this the carer copy? Was the method that patients who receive any help were only expected to return a carer questionnaire and not a patient questionnaire?

Response: Yes, this refers to the carer copy of the survey. In the Patient Information Sheet it was made clear that if anybody was helping the patient with medication, the patient was invited to pass to them the invitation letter, information sheet and survey labelled 'family member/friend or paid carer', provided they were happy for them to fill it. It was also made clear that it was ok for the patient and family member, friend or paid carer to provide different answers and for both of them to return their filled survey in the reply envelope to the research team.

Page 7 line 55 "...data were collected from the patient's questionnaire only" This reads as though both patient and caregiver invited to respond, but only one of these data used.

Response: Yes, that is correct. When both the patient and carer questionnaire were included in the reply envelope, only data from the patient questionnaire were used in the analysis. Carer's responses were used when the reply envelope included the caregiver's questionnaire only.

RESULTS

• Participant characteristics – "a high proportion of white participants" – provide percentage.

Response: The authors have now included a percentage for the proportion of white participants (79%), see page 10, line 15.

• It could have been useful to know whether respondents live alone or not, have formal home care or not, have cognitive impairment or aphasia if available.

Response: we agreed this extra information would have been helpful, though the study and ethic approval was not set up to collect these data, unfortunately.

• Table 1 – could add two rows, as well as those who receive help, also those with unmet needs, and those who missed meds – give that these three are the main arms of study findings.

Response: Thank you for this suggestion. Descriptive information of Patients with unmet needs and Patients who miss medications has now been added to Table 1, as suggested. Page 24

• Were all analyses conducted on the full sample of returned questionnaires? Was this n=596?

Response: Yes, analyses were conducted on the entire sample, n=596.

• Page 10 line 12 – why is the 55.7% not 56.3% i.e. 331/588 as per table 1?

Response: Thank you for this observation, which has allowed us to provide a more accurate figure. The total sample included in the analysis was 596. Therefore 331/596 is 55.5%, see page 11, line 3.

• Table 2 – why is 'n' missing for 3a, 4a, 5a?

Response: Thank you for highlighting these omitted values. We have now included all n values for question 3a, 4a and 5a in Table 2, page 25.

• Low return rate indicates poor acceptability of the survey, perhaps respondents felt the questionnaire didn't address their issues – is it possible to comment on or learn from any data about acceptability of the survey (eg in early testing, additional comments respondents may have written, or patterns in missing data)?

Response: The authors acknowledge that the low response rate is a limitation of this survey study. Response rate varied between top 53% in top and 16% in the poorest recruiting practices. No practice from Eastern England recruited less than 20% of patients, while 3 London practices did so. Nevertheless, other London practices recruitment was close to the average 35%, with the top recruiting practice showing a response rate of 40%. Therefore there was no clear pattern in missing data. Comments provided by respondents did not address the issue. We have added a paragraph mentioning a possible reason of our low response rate, i.e. older age and consequent difficulties with filling surveys. In fact, our participants' population is younger compared with the average age at stroke in UK. See page 13, lines 25-26.

- If needed to reduce word count maybe ORs, CIs and p values could be removed from the text as these are in the tables and the text adequately illustrates the main points.

Response: The authors thank the Reviewer for this suggestion. The authors have streamlined the written results to reduce repetition with the information provided in the Tables.

- An OR of 1.02 is reported as showing association (page 11 line 5), while a similar OR of 1.07 is reported as no association (page 11 line 19)

Response: We acknowledge this may appear odd, although we have checked again the statistical analyses and the results are confirmed.

DISCUSSION

- Page 12 line 7 – re taking a higher number of daily medications – keeping this depends on what you decide re difference between OR of 1.07 & 1.02.

Response: We identified statistically significant associations between number of different medicines and unmet needs (OR: 1.2 (1.1-1.3) $p < 0.001$) and between number of different medicines and missing medication (OR: 1.1 (1.0-1.1) $p = 0.008$). The results reported in Tables 3 and 4 and are accurate to the best of our knowledge according to the analysis we have undertaken.

- Page 12 line 11 & line 34 (also page 3 line 31), data is provided on those who are independent according to Barthel, but no data on those more dependent, making it difficult to be sure whether this study includes patients “severely affected by stroke”. One option could be including more dependent Barthel scorers in another column in table 1. A study method which is inherently biased against include those with cognitive impairment/aphasia (ie requires talking in a workshop or reading and writing the questionnaire), would need to be explicit about efforts to include them for this to be a representative stroke sample. If no efforts were taken this remains a useful and important survey of those who responded, but would be careful about wording around the diversity of this group unless demonstrated.

Response: This is an important point. During the 3 workshops to develop the questionnaire, the 12 caregivers were doing most of the talking on behalf of the patients they were looking after, who were moderately/severely impaired by stroke.

Within the survey, the description of patients moderately and severely dependent has now been added to Table 1. This data shows that the survey included data from this group of patients. The analysis has also incorporated categories of independence (i.e. Moderate dependence and Severe dependence for ADLs) and the associations with each category are reported in Tables 3 and 4.

- Page 12 line 31 – remove ‘actual’

Response: Thanks, the authors have removed ‘actual’ from page 13, line 18.

- Page 15 line 31 – insert full stop after ‘other patient groups’

Response: Thanks, the authors have inserted a full stop after ‘other patient groups’. Page 16, line 20

VERSION 2 – REVIEW

REVIEWER	Sureshkumar Kamalakannan
----------	--------------------------

	PHFI - India
REVIEW RETURNED	18-Dec-2017

GENERAL COMMENTS	The authors have responded, addressed and revised the manuscript significantly.
---

REVIEWER	Amanda Thrift Monash University, Australia
-----------------	---

REVIEW RETURNED	24-Dec-2017
-------------

GENERAL COMMENTS	The authors have addressed all of my concerns. Thank you.
---

REVIEWER	Rosie Shannon Bradford Institute for Health Research, UK
-----------------	---

REVIEW RETURNED	27-Dec-2017
-------------

GENERAL COMMENTS	Thank you for this resubmission. Author replies, amendments and explanations are appreciated and have contributed to an improved version of this article. Minor comments:  * It would be nice to know possible reasons for GP exclusions and proportion of each exclusion type. However, I appreciate the practicality of recruiting through busy practitioners who may be unable to pass this data on to the researchers. * Inviting both patient and carer to return their respective surveys but only using one in analysis would seem an issue of ethics – collecting data that was never intended to be used. Perhaps this is an issue of practicality. * I note that some OR figures have been rounded up/down and think this is now clearer (with regard to interpretation of significant/non significant associations). Could the authors also ensure the numbers in text match the table, for reader ease (eg OR of 1.07 is now rounded up in table to 1.1 but still reported in text as 1.07).
---

VERSION 2 – AUTHOR RESPONSE

Editorial requests:

- Along with your revised manuscript, please include a copy of the STROBE checklist indicating the page/line numbers of your manuscript where the relevant information can be found (<https://strobe-statement.org/index.php?id=strobe-home>)

Response: Thank you. We have now included a copy of the STROBE checklist indicating which page the relevant information can be found within the manuscript.

Reviewer(s)' Comments to Author:

Reviewer: 2

Reviewer Name: Sureshkumar Kamalakannan

Institution and Country: PHFI - India

Please state any competing interests or state 'None declared': None

Please leave your comments for the authors below The authors have responded, addressed and revised the manuscript significantly.

Reviewer: 3

Reviewer Name: Amanda Thrift

Institution and Country: Monash University, Australia

Please state any competing interests or state 'None declared': None declared

Please leave your comments for the authors below The authors have addressed all of my concerns. Thank you.

Reviewer: 4

Reviewer Name: Rosie Shannon

Institution and Country: Bradford Institute for Health Research, UK

Please state any competing interests or state 'None declared': None declared

Please leave your comments for the authors below Thank you for this resubmission. Author replies, amendments and explanations are appreciated and have contributed to an improved version of this article.

Minor comments:

* It would be nice to know possible reasons for GP exclusions and proportion of each exclusion type. However, I appreciate the practicality of recruiting through busy practitioners who may be unable to pass this data on to the researchers.

Response: We agree that the reasons for GP exclusions beside the actual study inclusion/exclusion criteria would have been interesting to look at, though we did not collect this data from GPs for practical reasons. We have added a sentence into the manuscript, clarifying this. See page 7 line 9-10.

* Inviting both patient and carer to return their respective surveys but only using one in analysis would seem an issue of ethics – collecting data that was never intended to be used. Perhaps this is an issue of practicality.

Response: Yes, this was an issue of practicality. The reason for sending a copy of the survey both to the participant and the caregiver, was to make sure patients with severe disabilities were included in the study, which was one of our goals, considering they represent an understudied population.

* I note that some OR figures have been rounded up/down and think this is now clearer (with regard to interpretation of significant/nonsignificant associations). Could the authors also ensure the numbers in text match the table, for reader ease (eg OR of 1.07 is now rounded up in table to 1.1 but still reported in text as 1.07).

Response: Thank you for this observation. We have now rounded up all numbers in the text to match those in the tables. See page 11, line 22 -25 and page 12 lines 9-13.